# Size Gynomimicry in the Sanmartinero Creole Bovine of the Colombian Orinoquia

**DOI:** 10.3390/vetsci11070304

**Published:** 2024-07-05

**Authors:** Arcesio Salamanca-Carreño, Pere M. Parés-Casanova, Mauricio Vélez-Terranova, Germán Martínez-Correal, David Eduardo Rangel-Pachón

**Affiliations:** 1Facultad de Medicina Veterinaria y Zootecnia, Universidad Cooperativa de Colombia, Villavicencio 500001, Colombia; 2Department de Bromatologia, Universitat Oberta de Catalunya, 08018 Catalonia, Spain; 3Facultad de Ciencias Agropecuarias, Universidad Nacional de Colombia sede Palmira, Palmira 763531, Colombia; 4Asociación de Criadores de Bovinos de Razas Criollas y Colombianas de los Llanos Orientales, Villavicencio 500001, Colombia

**Keywords:** anatomical structures, body size, morphology, sexual dimorphism

## Abstract

**Simple Summary:**

The external variations between animals of the same species but of different sex are called sexual dimorphism. Variations in size are called sexual size dimorphism. In the creole bovine, the phenomenon of sexual dimorphism has been little studied. The aim of this study was to establish whether sexual dimorphism appears in the Sanmartinero creole bovine, in the department of Meta, Colombia. A total of 21 linear variables were obtained using standard morphometric methods and live weight, from a sample of 94 animals (16 uncastrated males and 78 females) with an average age of 4.3 ± 1.4 and 4.2 ± 2.3 years were measured. Statistically significant differences were found between sexes (*p* = 0.033) and not by age and farms. The variables that most differentiated males from females were thoracic circumference, body length, height at the withers, height at the rump, dorso-sternal diameter, and horn length. The only variables that presented statistically significant differences were the height at the withers and the rump, with values biased toward males. More studies are needed to understand sexual size dimorphism in the Sanmartinero creole bovine.

**Abstract:**

Variations in the size of animals of the same species but of different sex are called sexual size dimorphism. The aim of this study was to compare the biometrics between males and females of the Sanmartinero creole bovine, of Colombia, to establish if sexual dimorphism appears in the breed. A total of 94 animals (16 uncastrated males and 78 females, average age of 4.3 ± 1.4 and 4.2 ± 2.3 years, respectively) from three different farms were measured. A total of 21 linear variables were obtained using standard morphometric methods and live weight. A one-way NPMANOVA was used to evaluate between sexes, ages, and farms, a Principal Component Analysis was used to detect the most discriminating variables, and a multivariate regression used age as an independent value. Statistically significant differences were reflected between sexes (*p* = 0.033) and not by age and farms. The variables that differentiated the most between males and females were those related to size (thoracic circumference, body length, dorso-sternal diameter, height at the withers, height at the rump, and horn length), variables that were biased toward males, although only the height at the withers and the rump were the ones that presented statistically significant differences.

## 1. Introduction

Body measurements provide explanatory information on morphometry in animals to determine the most appropriate structure according to their productive interest [1], and to demonstrate sexual differentiation [2]. The differences between sexes are of two types: those that directly involve the sexual organs from the animal’s birth and do not change during life (male and female); and those that develop when the animals reach reproductive age (e.g., size, horns, plumage, etc.) [3,4]. The last differences are called “secondary” sexual characters and are functionally related to display or fighting, and their variation is due to ecological factors [3,4]. When secondary sexual characters are selected and developed over others for their greatest reproductive success, it is called sexual selection [4].

Sexual dimorphism (SD) specifies the existence of phenotypic, morphological, physical, and physiological variations, unrelated to the sexual organs, between individuals of the same species but of different sex [5,6,7,8]. These external variations are manifested as dimorphism in size, body shape, shape and size of the appendages, integumentary characteristics, and coloration, and sexual differences in behavior [5,7,9,10]. Put another way, females and males can differ in size, shape, development of appendages (horns, teeth, feathers, or fins), and in the production of sounds and scents [11,12,13]. Body size variation occurs within and between related species, intraspecific and intrapopulation, and within a set of animals, allowing us to understand their adaptation and evolution to an environment [14,15].

The SD can be influenced by evolutionary restrictions, sexual selection, natural selection [11,16], and seasonal changes [17]. This trait is considered almost constantly associated with the action of sexual hormones and, therefore, with the formation of the gonads [18]. An essential mechanism of the variation in the size of the animal is the differences in body size between females and males, a phenomenon known as sexual size dimorphism (SSD) [13]. SSD is a widely distributed phenomenon in animals; however, it is enigmatic in terms of allometric relationships (Rensch’s rule, which interprets the relationship of the SSD with body size) [11,19,20,21]. The SSD is considered as an allometry of the sexual size, but there are still no explanations of the causes; however, some hypotheses say that it can be due to evolutionary restrictions, natural selection, or sexual selection [11,13].

Sex-specific differences, especially in general size and some body parts, are a recurring theme in the field of biology, and mainly focus on evolutionary issues in all species. In the case of bovines, males usually have larger body measurements than females, which determines a greater body volume [22]. The phenomenon of gynomimicry (the one by which males present an appearance similar to that of females), in particular, has been described in various domestic breeds, especially tropical ones [12,23].

In modern breeding programs, which select for economically important traits, sexual variation in traits is mitigated [24]. It can be assumed that in this way, sexual selection loses its significance. Since the mating decision is also made by humans when breeding domestic animals, sexual selection presumably plays a subordinate role. On the other hand, breeding programs evaluate important economic traits in males and females as a single trait, assuming a genetic correlation of “1”, which may not be true since genes can act differently in males and females [24].

The Sanmartinero creole bovine breed is typical of the ecosystem of the plains and highlands of the Meta department in the Colombian Orinoquia. It is believed to derive from the Spanish bovine from Extremadura, introduced in the 15th century by the Spanish conquerors. Since its introduction in America, it has lived in environments with extreme temperatures and has developed adaptive characteristics (e.g., reproductive efficiency and disease resistance) and ease of feeding with fibrous forages [25,26]. According to FAO parameters, its status is not at risk [27].

Its meat is of high quality and the milk contains kappa-casein B, a greater amount of fat, protein, and total solids. It presents productive attitudes (slaughter weight) as a pure breed and in crosses with Zebu Brahman. In crosses with Angus and with other creole cattle, such as the Blanco Orejinegro and the Romosinuano, it presents an excellent carcass performance [28].

Evidence of sexual dimorphism has been reported in some creole bovines such as the creole of Uruguay (chest width, thoracic perimeter, body length, rump width, cannon perimeter) [29], Pantaneiro creole (they only mention that there is sexual dimorphism) [30], Limonero creole (height of withers, body length, rump width and length, cannon perimeter, head width) [31], creole from Oaxaca (body length, thoracic perimeter, rump length and width [32], Casanare creole (head length and width) [33], and creole from Ecuador (head width, skull length, face length, height at withers, bicostal diameter, thoracic perimeter, cannon perimeter) [34]. There are some works on biometrics about the Sanmartinero creole bovine [25,26], and genetic parameters of pre-weaning growth [35]. However, to date, to the authors’ knowledge, there are no studies on sexual size dimorphism, e.g., considering globally all traits as an expression of size.

The hypothesis of the present study is to investigate whether sexual size dimorphism exists in this creole bovine. Therefore, the aim was to compare the biometrics between males and females of the Sanmartinero creole bovine to establish how the phenomenon of sexual dimorphism is expressed in the bovine.

## 2. Materials and Methods

### 2.1. Study Area

This study was carried out in the ecosystem of the plains and high plains in the department of Meta, in the Colombian Orinoquia (Figure 1) (Latitude: 01°36′29″ and 04°54′24″ N; Longitude 71°04′42″ and 74°54′09″ W). The regions belong to tropical humid and very humid tropical forest zones. Its topography is flat and undulating with an altitude of 350 m. The average environmental temperature is 26 °C; relative humidity of 71%; and average annual precipitation of 3100 mm [26].

### 2.2. Data Source

For the present study, 78 females and 16 uncastrated males were measured with an average age of 4.2 ± 2.3 and 4.3 ± 1.4 years for females and males, respectively, coming from three different farms in the department of Meta (Colombia) (Figure 1). The reduced number of males is because, according to information from farmers, the animals are sold at the time of weaning, so it is rare to find males on the farms.

The farms belong to breeders associated with the Association of Creole and Colombian Cattle Breeders of the Eastern Plains (ASOCRIOLLANOS, for its acronym in Spanish). On farms, direct mating with several bulls is used. The main food sources are native legumes and grasses. Mineralized salt is also supplemented [36].

The Sanmartinero creole bovine is defined as mesoline, eumetric, and orthoid (Figure 2). They are red animals, although there are some blacks, overos, browns, and isabelas. They have short ears, and the horns in cows are lyre-shaped; in bulls, they are directed forward in the shape of a crown. It has strong horns that enable it to repel the attack of feline predators. Given the extensive management to which it has been subjected and the prevailing environmental conditions in the ecosystem (extreme conditions of temperature and humidity), it has developed characteristics such as rusticity and the ability to live where foraging is scarce, and the availability of water spans long distances [26].

### 2.3. Morphometric Variables

A total of 21 linear body variables were obtained using standard morphometric methods (Table 1):

The variables were taken with a ruler and tape measure (Ovny, Inalmet, CO), and the age was obtained according to the information available in the record of each animal. Body weight was taken with a scale (Softgan Electronics SAS). The animals were immobilized in “brete” with a cement floor to facilitate the taking of measurements. The measurements were taken twice and were carried out during the rainy period (April to November). The graphic representation of the measurements taken is shown in Figure 3. The variable body weight was only considered as a statistical descriptor, not being included (since it is a value highly conditioned by dietary management) in subsequent multivariate analyses. Two trained students performed all measurements during field work.

### 2.4. Statistical Analysis

As some variables presented non-normal distributions, non-parametric tests were used. First, a one-way NPMANOVA (Non-Parametric Multivariate Analysis of Variance) was performed, using Mahalanobis distances and Bonferroni correction, to assess whether there were differences in the set of variables between sexes. Next, a two-way NPMANOVA was carried out, with 9.999 permutations, to detect possible differences between ages and farms. The Principal Component Analysis (PCA) was obtained from the var–covar matrix. The comparison of means was performed using a Mann–Whitney U test. Finally, with the variables with discharge values > 0.2, a multivariate regression was established for each sex, using age as the independent value and these variables as dependent values—in both cases, with the log-transformed values.

The statistical treatment was carried out with the statistical program PAST v. 2.17c [37], establishing the confidence level at 95%.

## 3. Results

Table 2 shows the main simple statistics for uncastrated males and females. The NPMANOVA reflected statistically significant differences between the sexes (*p* = 0.033). No statistically significant differences by age and farms were reflected (Table 3).

When applied to morphometric data, the PCA makes it possible to isolate the variance due to differences in size in the first axis, and in the subsequent axes, the variance is due to differences in shape. The combination of variables that had a greater impact on the grades obtained in this study was explained by the first two axes of the PCA, which explained 72.8% of the total variation observed (PC1 + PC2 = 61.13% + 11. 69%), showing an overlap of both sexes (Figure 4). All discharge values presented positive values, indicating that the differences between sexes are basically due to differences in size. The TC, BL, DD, HW, HR, and LHL were the variables that differentiated the most between males and females (discharge values > 0.29), indicating that the bulls are taller and have greater thoracic capacity, although without statistically significant differences, except for HW and HR, for which statistically significant differences between sexes are reflected (*p* = 0.00021, U = 256; *p* = 0.0029, *U* = 328, respectively).

In the multivariate regression, an isometry appears in the case of males (R^2^ = 0.0045; Wilk’s λ = 0.395; F_6.9_ = 2.292; *p* = 0.127), and a regression in the case of females (R^2^ = 0.172; Wilk’s λ = 0.529; F_6,70_ = 10.35; *p* << 0.0001).

## 4. Discussion

Morphological and phenotypic characteristics are commonly used to characterize animal breeds [38]. Selective pressure in the animal kingdom acts differently in each sex, generating external morphological variations or sexual dimorphism [4,39]. Sexual dimorphism is an important attribute in understanding animal adaptation to a social and natural ecosystem and is thought to be a result of sexual selection and natural selection [40,41].

The results of the present study show that males are taller and have greater thoracic capacity measured by their TC, BL, DD, and LHL, although these measures did not present significant statistical differences between sexes (*p* > 0.05). It could be deduced that males present an early interruption of their growth; that is, in biological terms, they do not fully develop all morphological characters, including secondary sexual ones. Studies carried out show that, in most higher mammals, sexual dimorphism is characterized mainly by males with a larger body size [42]. Body size is one of the most important characteristics in evolutionary and ecological terms since it correlates with aspects of its life history and is affected by biotic and abiotic factors [13]. In a study of nine zoometric variables of creole bovines from Panama, sexual dimorphism was reported only for the TC; however, the researchers did not report toward which sex it is biased [43].

The greater size in males could be favored by the sexual selection that occurs due to the male–male competition to obtain females, since the largest ones manage to dominate in the fights for females and thus achieve reproductive success and dominance of territory [44]; this behavior is common in polygamous mammals [45]. On the other hand, males tend to obtain more nutritional components to increase body size for alternative selection purposes [46].

The HW and HR reflected statistically significant differences between sexes biased toward males (*p* < 0.05), which expresses that the Sanmartinero creole bovine presents a sexual dimorphism of size explained by these two body measurements. These two measurements are not related to ethnological traits, so the differentiation may be associated with artificial breeding and selection, and not associated with ecological conditions of the foothills and high plains of the department of Meta. A previous study showed higher values for height at the withers in males and females, but without considering sexual dimorphism in size [25]. These results differ from those reported for creoles from the Ecuadorian coast, where females presented higher values for HW and HR, with marked sexual dimorphism only for HW (*p* < 0.05) [47]. On the other hand, sexual dimorphism could be explained by intersexual differences in reproductive functions and eating habits [48].

On the other hand, sexual differences in morphology may be due to intrasexual competition, which leads to strong sexual selection [40]. The sexual differentiation given by these two traits (in our case, HW and HR) could also be influenced by the competition for survival and reproduction, which can lead to the evolution in the increase in the body size of males [49,50]. In both sexes, the highest value was shown in HR, presenting an ascending dorsal lumbar line toward the rump, which is a characteristic that is reflected in animals with little artificial selection [51].

When comparing the results with other creole bovines, a sexual dimorphism in BL and TC is reported for the Oaxacan creole bovine, but biased toward females [32], results contrary to what was reported in the present study. In the criollo bovine of Santa Elena, Ecuador, sexual dimorphism was present in 9 of 14 variables analyzed, with greater variability for males [34]. In the Lojano creole bovine, sexual dimorphism was expressed in 10 of 16 body measurements studied, with higher values in males [52]. In the Limonero creole bovine, sexual dimorphism was evident in the hip perimeter and head width [31]. In the Casanare creole bovine, sexual dimorphism was found for head length and width, biased toward males [33]. In the Bolivian creole bovine, the differences were for head width [53], and in the Argentinian creole bovine it was found that males have larger body dimensions while females present wider rumps [54]. As can be seen, the data on sexual dimorphism in the Sanmartinero creole bovine contrast with the reports on other Ibero-American creole bovines.

Horn length (LHL) was slightly higher in males, which could be inferred as a lateralized adaptive characteristic for greater animal defense. This result is like that reported in creole bovines from Ecuador, where males also have length horns [47]. The size and conformation of the horns is a characteristic little influenced by the environment that generates important information on animal diversity [55]. In many groups of animals, it has been shown that the size of the horns is decisive in the animal’s defense and, consequently, in the reproductive success [44]. Ear width (LEW) was similar in males and females, while the length of the ear (LEL) was greater in males, which may indicate greater hearing capacity and possibly better temperature regulation for these. The DD was slightly higher in males.

Seasonal changes can produce sexual dimorphism, causing one of the two sexes to accelerate their sexual maturation to reproduce [17]. Other studies mention that males and females differ in traits measured at the end of the animal’s life, such as growth and weight characteristics [24]. Extensive animal management styles associated with low anthropogenic influence tend to reinforce sexual size dimorphism, which can be interpreted as integral variation between sexes [12]. Males and females can also exhibit morphological variations related to ecology because they live in different places or eat different foods [4,8,56]. Recent studies on ontogenetic and evolutionary development have made it possible to understand that group size, food, and the environment have driven the phenomenon of sexual dimorphism [41].

The main limitation of the present study is that the sample size of males was limited to 16 animals and the number of farms. However, we consider our results important for the Sanmartinero creole bovine breed since information is lacking about sexual size dimorphism. On the other hand, it is a breed that is competitive for the sustainable livestock development of the region given its adaptation for several generations. Future research can be done on the field of sexual dimorphism from an ecological point of view. In fact, gynomimicry has not been extensively studied in domestic breeds. It opens the door to future research taking into account the interaction between animal and environment, normally stronger than husbandry practices among creole breeds managed under semi-extensive conditions, as Sanmartinero is. For future selective planning, it will be important to consider gynomimicry as an adaptive trait in order to avoid a selection toward “masculinized” bulls.

## 5. Conclusions

In this study, Sanmartinero creole males presented an isometry with higher zoometric values than females, and they have greater thoracic capacity. The sexual size dimorphism was only expressed in two body measurements (height at the withers and height at the rump) that do not represent ethnological characters. Therefore, ecological studies are needed to understand the sexual size dimorphism in these bovines.

## Figures and Tables

**Figure 1 vetsci-11-00304-f001:**
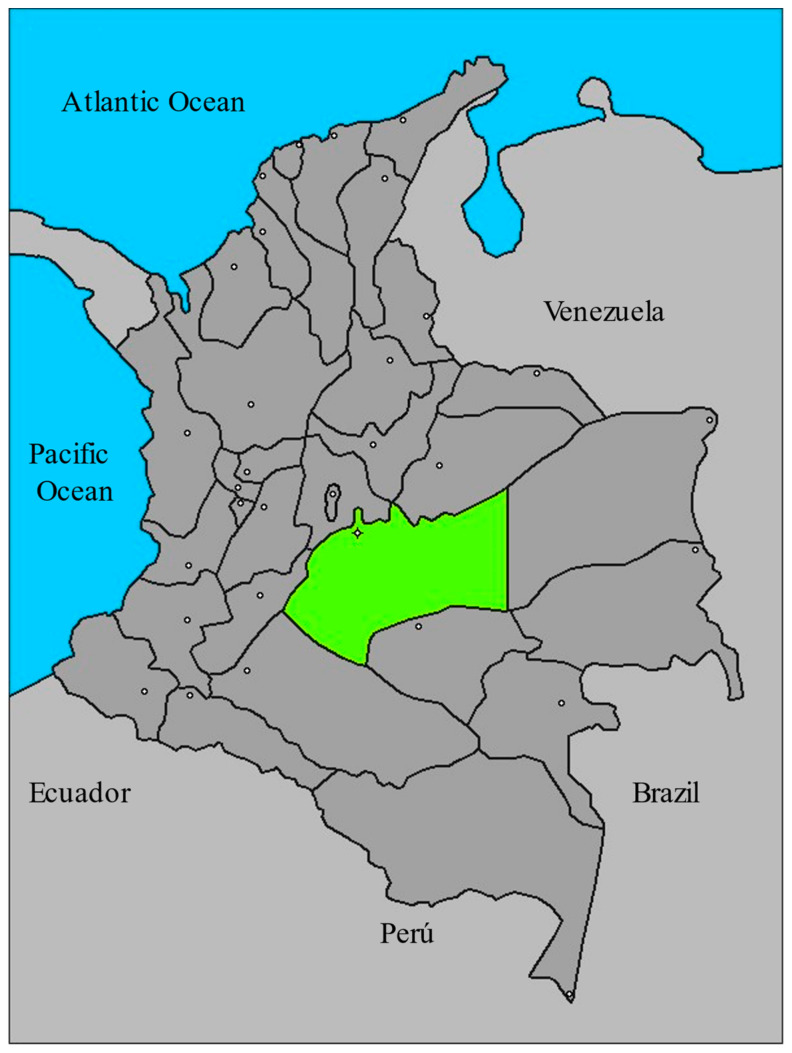
Map of Colombia. Green color: department of Meta in the Colombian Orinoquia.

**Figure 2 vetsci-11-00304-f002:**
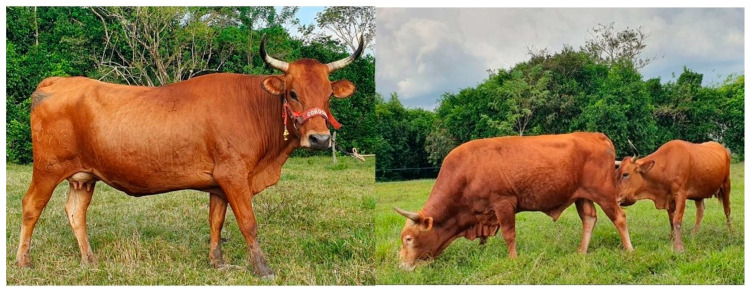
Female and male Sanmartinero creole bovine, a bovine that adapts to the environmental conditions of the foothills of the plains and high plains of the Meta department, Colombian Orinoquia. Photograph taken at the Punta Hermosa farm by the owner.

**Figure 3 vetsci-11-00304-f003:**
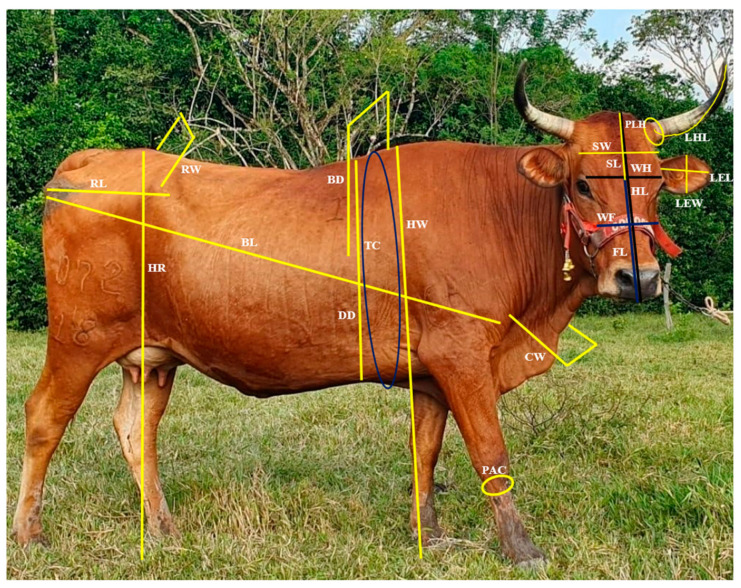
Graphic representation with arrows of the body measurements taken for the Sanmartinero creole bovine. BL: body length; TC: thoracic circumference; HW: height at the withers; BD: bicostal diameter; DD: dorso-sternal diameter; CW: chest width; PAC: perimeter of the anterior cannon; HR: height to the rump; RW: rump width; RL: rump length; WH: width of the head; HL: head length; WF: width of face; FL: face length; SW: skull width; SL: skull length; PLH: perimeter of left horn; LHL: left horn length; LEL: left ear length; LEW: left ear width. Photograph taken at the Punta Hermosa farm by the owner.

**Figure 4 vetsci-11-00304-f004:**
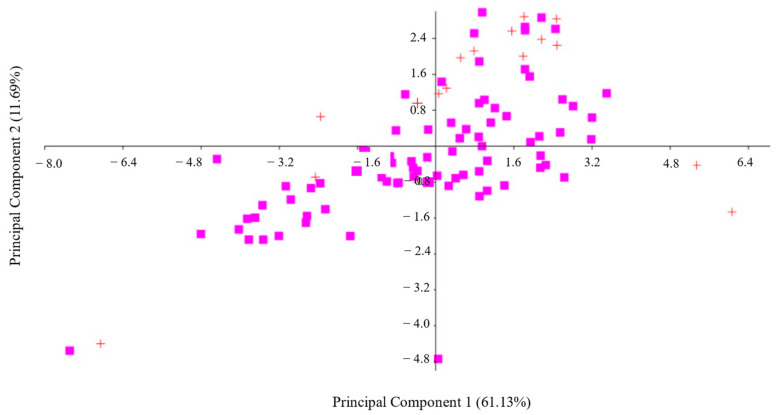
Analysis of Principal Components for uncastrated males (*n* = 16 +) and females (*n* = 78 ■) of the Sanmartinero creole bovine for the 21 variables studied. The first two axes of the PCA explained 72.8% of the total variation observed (PC1 + PC2 = 61.13% + 11.69%), showing an overlap between both sexes.

**Table 1 vetsci-11-00304-t001:** Linear body variables taken for the Sanmartinero creole bovine (includes body weight).

Body Variables	Description
Body length (BL)	Distance between the most cranial and lateral point of the humeral joint and the most caudal ilio-ischial point
Thoracic circumference (TC)	Measurement of the girth around the chest, passing through the withers and the sternum
Height at the withers (HW)	Distance from the floor to the highest point of the withers—interscapular region, 3rd, and 4th spinous process of the thoracic vertebrae
Bicostal diameter (BD)	Distance between both costal planes, taking as reference the limits of the costal region, at the level of the arch of the 5th rib
Dorso-sternal diameter (DD)	Distance between the lowest point of the withers and the point of greatest curvature of the sternum
Chest width (CW)	Distance between the most cranial and lateral points of the encounters
Perimeter of the anterior cannon (PAC)	Perimeter in the narrowest part of the metacarpal bone- in its middle and upper thirds
Height to the rump (HR)	Distance from the floor to the point of union of the loins with the rump
Rump width (RW)	Distance between the most lateral and cranial points of the coxal tuberosity
Rump length (RL)	Distance between the most protruding lateral point of the coxal tuberosity and the most caudal ilio-ischiatic point
Width of the head (WH)	Distance between the most prominent points of the zygomatic arches
Head length (HL)	Distance between the most culminating point of the occipital -nape- and the most rostral -anterior- of the maxillary lip
Width of face (WF)	Straight length between the most lateral points of the malar ridges
Face length (FL)	Straight length between the midpoint of the fronto-nasal junction and the most rostral point of the maxillary lip
Skull width (SW)	Distance between the points immediately superior to the coronoid process of the mandibular rami
Skull length (SL)	Distance between the most prominent point of the nape-Occipital- and the midpoint of the fronto-nasal junction
Perimeter of left horn (PLH)	Measurement around the base of the crown
Left horn length (LHL)	Distance from the base of the crown to the apex of the horn
Left ear length (LEL)	Distance from the base of the ear insertion to the vertex
Left ear width (LEW)	Distance from the midpoint of the cranial border to the midpoint of the caudal border
Body weight (BW)	Taken with a scale

**Table 2 vetsci-11-00304-t002:** Main simple statistics for uncastrated males (*n* = 16) and females (*n* = 78) of the Sanmartinero creole bovine, for the 21 variables studied (see in the text). Measurements in cm, except for BW, in kg.

Quantitative Trait	Males					Females				
	Min	Max	Mean	SD	CV (%)	Min	Max	Mean	SD	CV (%)
BL	96.0	170.0	143.9	19.2	13.4	95.0	162.0	138.4	14.2	10.2
TC	120.0	218.0	170.6	23.0	13.5	112.0	195.0	165.6	18.1	10.9
HW	82.0	134.0	123.5 a	12.8	10.3	82.0	136.0	115.6 b	8.5	7.3
BD	32.0	60.0	45.8	6.4	14.0	29.0	65.0	44.7	7.9	17.7
DD	40.0	83.0	61.3	9.9	16.2	32.0	80.0	59.6	10.5	17.6
CW	26.0	87.0	42.4	15.3	36.0	30.0	47.0	34.6	6.7	19.4
PAC	18.5	39.0	23.7	6.1	25.8	15.0	38.0	26.8	4.2	15.6
HR	87.0	136.0	126.9 a	12.5	9.8	12.0	138.0	119.8 b	15.4	12.9
RW	26.0	50.0	41.0	6.9	16.8	35.0	58.0	42.5	8.1	19.2
RL	30.0	53.0	45.3	6.6	14.7	30.0	60.0	45.1	5.7	12.7
WH	15.0	29.0	22.9	2.8	12.1	15.0	46.0	20.3	3.6	17.6
HL	36.0	57.0	50.8	6.8	13.4	23.0	56.0	49.1	5.2	10.7
WF	7.0	20.0	15.5	3.2	20.7	7.0	21.0	11.7	2.7	23.5
FL	21.0	36.0	30.7	4.6	15.0	3.0	36.0	27.8	5.0	17.9
SW	12.0	26.0	20.1	3.4	16.7	15.0	29.0	21.3	2.1	10.0
SL	16.0	27.0	21.1	2.9	13.7	14.0	27.0	21.6	2.4	11.3
PLH	9.0	31.0	21.3	5.2	24.5	6.0	23.0	18.1	2.9	15.9
LHL	4.0	33.0	25.4	7.3	28.7	3.0	51.0	23.4	9.2	39.4
LEL	14.0	23.0	18.6	2.3	12.3	7.0	23.0	16.1	2.0	12.7
LEW	9.5	12.0	10.4	0.7	6.7	8.0	14.0	10.5	1.0	9.5
BW	142.0	788.0	404.8	153.0	37.8	117.0	556.0	366.9	99.6	27.1

Min: minimum; Max: maximum; SD: standard deviation; CV: coefficient of variation; BL: body length; TC: thoracic circumference; HW: height at the withers; BD: bicostal diameter; DD: dorso-sternal diameter; CW: chest width; PAC: perimeter of the anterior cannon; HR: height to the rump; RW: rump width; RL: rump length; WH: width of the head; HL: head length; WF: width of face; FL: face length; SW: skull width; SL: skull length; PLH: perimeter of left horn; LHL: left horn length; LEL: left ear length; LEW: left ear width; BW: body weight. Different letters in the same row differ statistically (*p* < 0.05).

**Table 3 vetsci-11-00304-t003:** Two-way NPMANOVA (Non-Parametric Multivariate Analysis of Variance), using Bonferroni’s correlation and correction distances, with 9999 permutations, according to farm and sex for all the studied animals (*n* = 94). No statistically significant differences were reflected either by age or by farms (*p* > 0.05).

Source	Sum of Squares	Degrees of Freedom	Mean Square	F	*p*
Farm	0.001	2.000	0.001	0.825	0.168
Age	0.007	9.000	0.001	0.861	0.123
Interaction	−0.029	18.000	−0.002	−1.827	0.396
Residual	0.056	63.000	0.001		
Total	0.035	92.000			

## Data Availability

Data are available upon reasonable request to the second author.

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
