# Peer review of "Size Gynomimicry in the Sanmartinero Creole Bovine of the Colombian Orinoquia"

_vetsci, 2024, doi:10.3390/vetsci11070304_

Round 1

Reviewer 1 Report

Comments and Suggestions for Authors

Brief summary

The manuscript submitted for peer review in Veterinary Science by Salamanca-Carreño et al. titled “Size gynomimicry in the Sanmartinero creole bovine of the Colombian Orinoquia”, aimed to study sexual size dimorphism in a local Colombian creole cattle breed. The authors registered several morphometric parameters and compared them between sexes. The authors performed a Multivariate Analysis of Variance and a principal component analysis, ultimately tried to fit a multivariate regression model. Only two of the 21 measured parameters showed significant differences between sexes.

General concept comments

Strengths:

  1. Experimental design and response criteria are easy to understand and interpret.
  2. The manuscript includes a detailed introduction, which can be interesting to a more curious reader.

Weaknesses:

  1. Although detailed, the introduction, and the material and methods are very long and can be shortened if the manuscript is accepted.

2.    According to iThenticate, there is 11% similarity with the previously published article in Animals by the same authors: Early Cannon Development in Females of the “Sanmartinero Creole Bovine Breed”. Some of the similarities correspond to almost full sections, namely section 2.1 Study Area and some parts of 2.3 Data Source. These sections should be re-written and summarized.

  1. One of the main issues in this essay is the sample size. In such a simple experimental procedure, a considerable sample size would be expected. The study is based on values collected from 16 males and 78 females. But the fact that these animals belong to only three farms is even more worrying. Many small-scale pure-breed suckler herds end up having a genetic bottleneck derived from the fact that the majority of replacement heifers are the result of a limited genetic male pool. Therefore, this study might not reflect adequately the whole breed population.
  2. This study does not bring any new insight to veterinary or animal science since the experiment results only apply to this specific breed. Thus, its global impact is neglectable.

Specific comments:

  • Paragraph 89-97: unnecessary information. It is not relevant to the essay.
  • Paragraph 98-104: Natural selection is the process of organisms with certain, more favourable characteristics tend to reproduce more successfully, in nature. The modification/evolutionary process described is a result of selective breeding, not natural selection. In domesticated species, even under low anthropogenic influence survivability don´t depend solely in natural conditions. Assuming that genetic improvement programs were rudimentary five centuries ago, the entire article is summed up in the authors' curiosity to understand whether these human-oriented and random modifications created sexual size dimorphism.
  • Line 151: What is mineralized salt? Mineral licks/blocks?
  • Section 2.3. The morphometric variables list should be converted to a table.
  • Line 287-289: deductions/hypothesis should be discussed in the discussion section, not in the results.
  • Line 327-329: decontextualized. What is the point?
  • Paragraph 362-366. See comments on paragraph 98-104. I don’t think authors should try to explain any differences found with natural evolution/selection. Therefore, that paragraph should be eliminated along with any other references to natural evolutionary biology hypothesis.
Comments on the Quality of English Language

The English language used is acceptable although moderate editing is suggested, particularly in certain paragraphs (e.g.: 89-96 paragraph and 106-112 paragraph).

Author Response

Thank you very much for reviewing our manuscript. We greatly appreciate all your review for your complimentary comments and suggestions. We have carried out the changes that you have been suggested and revised the manuscript accordingly.

Please find attached a point-by-point response to each reviewer’s concerns. We hope that you find our responses satisfactory and that the manuscript will be now acceptable for  publication.

Sincerely,

 First reviewer’s responses

Comment

Brief summary

The manuscript submitted for peer review in Veterinary Science by Salamanca-Carreño et al. titled “Size gynomimicry in the Sanmartinero creole bovine of the Colombian Orinoquia”, aimed to study sexual size dimorphism in a local Colombian creole cattle breed. The authors registered several morphometric parameters and compared them between sexes. The authors performed a Multivariate Analysis of Variance and a principal component analysis, ultimately tried to fit a multivariate regression model. Only two of the 21 measured parameters showed significant differences between sexes.

 General concept comments

Strengths:

  1. Experimental design and response criteria are easy to understand and interpret.
  2. The manuscript includes a detailed introduction, which can be interesting to a more curious reader.

Weaknesses:

  1. Although detailed, the introduction, and the material and methods are very long and can be shortened if the manuscript is accepted.
  1. According to iThenticate, there is 11% similarity with the previously published article in Animals by the same authors: “Early Cannon Development in Females of the “Sanmartinero Creole Bovine Breed”. Some of the similarities correspond to almost full sections, namely section 2.1 Study Area and some parts of 2.3 Data Source. These sections should be re-written and summarized.

Response: corrected in text.

 Comment

  1. One of the main issues in this essay is the sample size. In such a simple experimental procedure, a considerable sample size would be expected. The study is based on values collected from 16 males and 78 females. But the fact that these animals belong to only three farms is even more worrying. Many small-scale fullblood suckler herds end up having a genetic bottleneck derived from the fact that the majority of replacement heifers are the result of a limited genetic male pool. Therefore, this study might not reflect adequately the whole breed population.

Response: okay. We have written in “Materials and Methods” section and at the end of the discussion the following observations: “The reduced number of males is because that, according to information from farmers, the animals are sold at the time of weaning, so it is rare to find males on the farms”. “The main limitation of the present study is that the sample size of males was limited to 16 animals and the number of farms. However, we consider our results important for the Sanmartinero creole bovine breed since information is lacking about sexual size dimorphism”.

 Comment

  1. This study does not bring any new insight to veterinary or animal science since the experiment results only apply to this specific breed. Thus, its global impact is neglectable.

Response: we consider that it is a valuable contribution for our creole bovine, especially Colombian creole bovine since there is no information in this regard (Sanmartinero creole bovine case). On the other hand, their global impact is not negligible, since many readers who may not know creole bovine may have knowledge of them. At the end of the text, it reads: However, we consider our results important for the Sanmartinero creole bovine breed since information is lacking about sexual size dimorphism. This is a starting point for future research.”

 Comment

Specific comments:

  • Paragraph 89-97: unnecessary information. It is not relevant to the essay.

Response. Corrected, removed from text.

  • Paragraph 98-104: Natural selection is the process of organisms with certain, more favourable characteristics tend to reproduce more successfully, in nature. The modification/evolutionary process described is a result of selective breeding, not natural selection. In domesticated species, even under low anthropogenic influence survivability don´t depend solely in natural conditions. Assuming that genetic improvement programs were rudimentary five centuries ago, the entire article is summed up in the authors' curiosity to understand whether these human-oriented and random modifications created sexual size dimorphism.

Response: a selective breeding (artificial selection) cannot be neglected: during centuries, cattle breeders have been selecting individual animals with desirable characteristics (such as meat and milk). Best animals were then bred to produce offspring. But natural selection on creole breeds must be considered also of “high impact” among this breed, which has been traditionally managed under extensive conditions. In other word, natural selection have selected adaptative natural characteristics (such disease resistance and extreme climate periods) over desirable traits. Our study is based on a creole bovine, that is, those bovines that were introduced to America in the 15th century and that have had a natural selection process for more than 500 years in different regions of America. I any case, it was corrected in the text.

Comment

  • Line 151: What is mineralized salt? Mineral licks/blocks?

Response: a mineralized salt is a mixture of Sodium Chloride (White Salt), Calcium and Phosphorus, and other minerals. Mineralized salt is used for animals to lick.

 Comment

  • Section 2.3. The morphometric variables list should be converted to a table.

Response: corrected, it became a table

Comment

  • Line 287-289: deductions/hypothesis should be discussed in the discussion section, not in the results.

Response. They have been corrected and moved on to the “Discussion” section.

Comment

  • Line 327-329: decontextualized. What is the point?

Response: it was deleted.

Comment

  • Paragraph 362-366. See comments on paragraph 98-104. I don’t think authors should try to explain any differences found with natural evolution/selection. Therefore, that paragraph should be eliminated along with any other references to natural evolutionary biology hypothesis.

Response: it has been removed.

Comment

The English language used is acceptable although moderate editing is suggested, particularly in certain paragraphs (e.g.: 89-96 paragraph and 106-112 paragraph).

Response: corrected.

Reviewer 2 Report

Comments and Suggestions for Authors

Dear Authors,

I have read your paper carefully and I find it interesting. I have some minor remarks and suggestions for your paper, listed below.

 Lines 114-119. You mention that sexual dimorphism in creole bovines has been reported. The papers you refer to are written in Spanish thus are not easily accessible for majority of scientific community. I suggest to shortly review what kind of dimorphism has been found by previous authors (body shape, horn size etc?).

Lines 228-241 (statistical analysis) – please provide justification for using nonparametric MANOVA instead of parametric one.

Discussion – The pattern of sexual dimorphism and sexual size dimorphism may be altered in the domestic animals, since, as you wrote, it is the farmer who makes decision which animal breeds. However, in the domestic lines, very common pattern is pedomorphosis – animals are getting mature retaining some juvenile characters. Please address this shortly in you paper (in the context of less pronounced sexual dimorphism).

Also, please address to sexual shape dimorphism and allometric patterns of the dimorphism you found in your study. There are interesting pattern of scaling the sexual dimorphism, that shows the larger sex is not just overgrown version of the smaller one, and it has serious implications for several aspects of its biology; see the papers below:

Borczyk B. 2023. Sexual dimorphism in skull size and shape of Laticauda colubrina (Serpentes: Elapidae). PeerJ 11:e16266 DOI 10.7717/peerj.16266

Wilson et al. 2022. Sex differences in allometry for phenotypic traits in mice indicate that females are not scaled males. Nature Communications 13:7502 DOI 10.1038/s41467-022-35266-6.

There are also some minor technical issues:

Lines 89-93, 106-111 and all references are written in different fonts.

Figure 3. The acronyms on that picture are very small and should be more contrast.

Figure 4. There are two axes named “Componente 1” and “Componente 2”. That should be PC1 and PC2 respectively (or Principal Component 1 and Principal Component 2). Also, the position of the “Componente 1” is slightly misleading, since at first sight it looks like it apply to the y-axe (the Componente 2).

Further – there is one outlier observation – Principal Component Analysis is sensitive to such outlying observations. Please address to this observation. What is “wrong” with this female? Please make sure, that the raw data are correct, since some trait(s) must weight more for this specimen. Perhaps it is so atypical one, that it could be justified to remove it from the data set?

Line 361: The sentence “(…) driven the phenomenon of sexual [44].” lack something. Do you mean “sexual dimorphism”?

References: Please use italics for species and generic names. Moreover, in some cases you provide full journal name and in others abbreviated name. Please make it uniform.

Author Response

Thank you very much for reviewing our manuscript. We greatly appreciate all your review for your complimentary comments and suggestions. We have carried out the changes that you have been suggested and revised the manuscript accordingly.

Please find attached a point-by-point response to each reviewer’s concerns. We hope that you find our responses satisfactory and that the manuscript will be now acceptable for publication.

Sincerely,

Second reviewer’s responses

 Comment

Dear Authors,

I have read your paper carefully and I find it interesting. I have some minor remarks and suggestions for your paper, listed below.

 Lines 114-119. You mention that sexual dimorphism in creole bovines has been reported. The papers you refer to are written in Spanish thus are not easily accessible for majority of scientific community. I suggest to shortly review what kind of dimorphism has been found by previous authors (body shape, horn size etc?).

Response: Corrected in text.

 Comment

Lines 228-241 (statistical analysis) – please provide justification for using nonparametric MANOVA instead of parametric one.

Response: Corrected in text.

 Comment

Discussion – The pattern of sexual dimorphism and sexual size dimorphism may be altered in the domestic animals, since, as you wrote, it is the farmer who makes decision which animal breeds. However, in the domestic lines, very common pattern is pedomorphosis – animals are getting mature retaining some juvenile characters. Please address this shortly in you paper (in the context of less pronounced sexual dimorphism).

Also, please address to sexual shape dimorphism and allometric patterns of the dimorphism you found in your study. There are interesting pattern of scaling the sexual dimorphism, that shows the larger sex is not just overgrown version of the smaller one, and it has serious implications for several aspects of its biology; see the papers below:

Borczyk B. 2023. Sexual dimorphism in skull size and shape of Laticauda colubrina (Serpentes: Elapidae). PeerJ 11:e16266 DOI 10.7717/peerj.16266

Wilson et al. 2022. Sex differences in allometry for phenotypic traits in mice indicate that females are not scaled males. Nature Communications 13:7502 DOI 10.1038/s41467-022-35266-6.

 Response.  Our study is in the Sanmartinero creole cattle (Bos taurus) (we consider a first study on the issue) and the two documents correspond to serpent’s and mice. However, we considered his recommendation and added the phrase:

 “On the other hand, sexual dimorphism could be explained by intersexual differences in reproductive functions in eating habits” (Ref. 48).

Comment

There are also some minor technical issues:

Lines 89-93, 106-111 and all references are written in different fonts.

Response: Corrected in text.

Comment

Figure 3. The acronyms on that picture are very small and should be more contrast.

Response: The acronyms in the image were corrected.

 Comment

Figure 4. There are two axes named “Componente 1” and “Componente 2”. That should be PC1 and PC2 respectively (or Principal Component 1 and Principal Component 2). Also, the position of the “Componente 1” is slightly misleading, since at first sight it looks like it apply to the y-axe (the Componente 2).

Further – there is one outlier observation – Principal Component Analysis is sensitive to such outlying observations. Please address to this observation. What is “wrong” with this female? Please make sure, that the raw data are correct, since some trait(s) must weight more for this specimen. Perhaps it is so atypical one, that it could be justified to remove it from the data set?

Response. The figure was corrected.

Comment

Line 361: The sentence “(…) driven the phenomenon of sexual [44].” lack something. Do you mean “sexual dimorphism”?

Response. Corrected in text.

Comment

References: Please use italics for species and generic names. Moreover, in some cases you provide full journal name and in others abbreviated name. Please make it uniform.

Response. Corrected

Round 2

Reviewer 1 Report

Comments and Suggestions for Authors

Dear authors,

I appreciate your quick response. Your comments and improvements have been helpful to a more robust and clear presentation of your research.

I have no further specific comments to make. Still, I would like the authors to explain me your response to comment number 4 quoted below:

This study does not bring any new insight to veterinary or animal science since the experiment results only apply to this specific breed. Thus, its global impact is neglectable.

2

Response: we consider that it is a valuable contribution for our creole bovine, especially Colombian creole bovine since there is no information in this regard (Sanmartinero creole bovine case). On the other hand, their global impact is not negligible, since many readers who may not know creole bovine may have knowledge of them. At the end of the text, it reads: “However, we consider our results important for the Sanmartinero creole bovine breed since information is lacking about sexual size dimorphism. This is a starting point for future research.”

Ok. But can you be a little more specific? What sort of future research is now possible with your study?
Could you please specify any practical solutions or improvements that your research will bring to the producers of Sanmartinero Creole Bovine?

Comments on the Quality of English Language

The English language has improved, particularly in the mentioned sections.

Author Response

First reviewer’s responses Round2

 We greatly appreciate all your review for your complimentary comments and suggestions. We have carried out the changes that you have been suggested and revised the manuscript accordingly.

 Comment

Dear authors,

I appreciate your quick response. Your comments and improvements have been helpful to a more robust and clear presentation of your research.

I have no further specific comments to make. Still, I would like the authors to explain me your response to comment number 4 quoted below:

This study does not bring any new insight to veterinary or animal science since the experiment results only apply to this specific breed. Thus, its global impact is neglectable.

2

Response: we consider that it is a valuable contribution for our creole bovine, especially Colombian creole bovine since there is no information in this regard (Sanmartinero creole bovine case). On the other hand, their global impact is not negligible, since many readers who may not know creole bovine may have knowledge of them. At the end of the text, it reads: “However, we consider our results important for the Sanmartinero creole bovine breed since information is lacking about sexual size dimorphism. This is a starting point for future research.”

Ok. But can you be a little more specific? What sort of future research is now possible with your study?
Could you please specify any practical solutions or improvements that your research will bring to the producers of Sanmartinero Creole Bovine?

Response: Future research can be done on the field of sexual dimorphism from an ecological point of view. In fact, gynomimicry has not been extensively studied in domestic breeds. It opens the door to future researches taking into account the interaction animal-environment, normally stronger than husbandry practices among creole breeds managed under semi-extensive conditions, as Sanmartinero is. For a future selective planning, it will be important to consider gynomimicry as adaptative trait, in order to avoid a selection towards “masculinized” bulls. It was included in the text.
